Combining environmental suitability and population abundances to evaluate the invasive potential of the tunicate Ciona intestinalis along the temperate South American coast

Januario Stella M. 1 2 stella.januario@uss.cl
Estay Sergio A. 3 4
Labra Fabio A. 5
Lima Mauricio 2 4
1 Departamento Ciencias Biologicas y Químicas, Facultad de Ciencias, Universidad San Sebastián , Valdivia , Chile
2 Departamento de Ecología, Facultad de Ciencias Biológicas, Pontificia Universidad Católica de Chile , Santiago , Chile
3 Instituto Ciencias Ambientales y Evolutivas, Facultad de Ciencias, Universidad Austral de Chile , Valdivia , Chile
4 Center of Applied Ecology and Sustainability (CAPES), Facultad de Ciencias Biológicas, Pontificia Universidad Católica de Chile , Santiago , Chile
5 Centro de Investigación e Innovación para el Cambio Climático, Facultad de Ciencias, Universidad Santo Tomas , Santiago Region Metropolitana , Chile
Castilho Rita
Electronic publication date: 2015 Oct 27
Publication date: 2015
Volume: 3
Electronic Location ID: e1357
Received 2015 Jul 21; Accepted 2015 Oct 7
Copyright: © 2015 Januario et al.
Copyright year: 2015
Copyright holder: Januario et al.
License: This is an open access article distributed under the terms of the Creative Commons Attribution License, which permits unrestricted use, distribution, reproduction and adaptation in any medium and for any purpose provided that it is properly attributed. For attribution, the original author(s), title, publication source (PeerJ) and either DOI or URL of the article must be cited.
License URL: https://creativecommons.org/licenses/by/4.0/

Keywords: Aquaculture, Invasion risk, Population abundance, Species distribution, Suitability index

Funding: FONDECYT # 3130373 CAPES-CONICYT FB-0002 This research was supported by a post-doctoral grant FONDECYT # 3130373 to SMJ, and the CAPES-CONICYT grant FB-0002 line 4 to ML and SE. The funders had no role in study design, data collection and analysis, decision to publish, or preparation of the manuscript.

==============================
The tunicate Ciona intestinalis is an opportunistic invader with high potential for causing economic losses in aquaculture centers. Recent phylogenetic and population genetic analysis support the existence of a genetic complex described as C. intestinalis with two main dominant species (sp A and B) occurring worldwide. In Chile, the species has been observed around 30°S of latitude, but no official reports exist for the presence of C. intestinalis in southern regions (above 40°S), where most of the mollusk aquaculture centers are located. Here, we used occurrences from multiple invaded regions and extensive field sampling to model and validate the environmental conditions that allow the species to persist and to find the geographic areas with the most suitable environmental conditions for the spread of C. intestinalis in the Chilean coast. By studying the potential expansion of C. intestinalis southward in the Chilean Coast, we aimed to provide valuable information that might help the development of control plans before the species becomes a significant problem, especially above 40°S. Our results highlight that, by using portions of the habitat that are apparently distinguishable, the species seem to be not only genetically distinct, but ecologically distinct as well. The two regional models fitted for sp A and for sp B showed disagreement on which sections of Chilean coastline are considered more suitable for these species. While the model for sp A identifies moderately to highly suitable areas between 30° and 40°S, the model for sp B classifies the areas around 45°S as the most appropriate. Data from field sampling show a positive linear relationship between density of C. intestinalis and the index of suitability for sp A in aquaculture centers. Understanding the relation of the distinct species with the surrounding environment provided valuable insights about probable routes of dispersion in Chile, especially into those areas considered suitable for aquaculture activities but where the species has not yet been recorded. We discuss the implications of our findings as a useful tool to anticipate the invasion of such harmful invasive species with regard to the most relevant environmental variables.

Introduction

Marine invertebrates are amongst the species with the highest potential for invasion and damage (Capinha, Anastácio & Tenedório, 2012; Lee et al., 2008; Robinson et al., 2011). Most of the incursions of invasive species in coastal areas are nowadays inevitable due to the worldwide traffic of marine vessels (Ramsay et al., 2008; Ashelby, Johnson & De Grave, 2013), and the co-transference of organisms during importation of commercially exploited species for stocking or aquaculture purposes (Locke & Hanson, 2009). Many of these opportunist species take advantage of human activity to extend their distributions, and are associated with aquaculture centers, causing large damage to both cultures and natural environments. This is the case of the tunicate Ciona intestinalis (Karayucel, 1997; Hecht & Heasman, 1999; Uribe & Etchepare, 2002), a sessile filter feeder that lives in dense aggregations in enclosed or semi-protected marine embayments (Carver, Mallet & Vercaemer, 2006). Many of its life history traits make this species a successful invader, including its rapid growth rates (20 mm/month), early maturation (8–10 weeks) and high reproductive output (>10,000 eggs/individual). In addition, it exhibits wide environmental tolerance, especially temperature (Carver, Chisholm & Mallet, 2003). Across its native range (North Atlantic) it is considered a dominant competitor in benthic communities, while in its exotic range it occurs as an opportunistic fouling organism on artificial substrates in harbors or in association with aquaculture equipment (Carver, Mallet & Vercaemer, 2006).

Recently, it has been discovered that the taxon C. intestinalis actually corresponds to a genetic complex of 2–4 species (Suzuki, Nishikawa & Bird, 2005; Iannelli et al., 2007; Zhan, Macisaac & Cristescu, 2010). Two of them, the species A and B are the most common forms, with the widest geographic distribution (Zhan, Macisaac & Cristescu, 2010). Both sp A and sp B are distributed worldwide: sp A has invaded the Pacific Ocean, the Mediterranean Sea, Australia and South Africa, while sp B occupies Northern Europe, including the British coastline, as well as the east coast of North America and Canada. The two remaining species, C and D, are rare, and remain restricted to small areas in the Mediterranean and Black Sea, respectively (Zhan, Macisaac & Cristescu, 2010). Recently, it has been proposed that sp A actually correspond to C. robusta and sp B to the original C. intestinalis (Brunetti et al., 2015; Pennati et al., 2015). Although there have been efforts to use phenotypic traits such as body color, pigmentation at the distal end of the siphons and the presence or absence of tubercles on the sides of the siphons to facilitate the identification of these different species in the field (Sato, Satoh & Bishop, 2012; Brunetti et al., 2015; Pennati et al., 2015), it is likely that available information regarding the distribution of the species is a mix of records of the whole genetic complex. For this reason, in this study we will maintain the sp A and sp B classification, but keeping in mind the new proposal taxonomy.

Over the last few decades population outbreaks have been observed at many sites around the world including South Africa (Hecht & Heasman, 1999), Scotland (Karayucel, 1997), and Chile (Uribe & Etchepare, 2002). As a result, the species has become a serious biofouling problem to the marine aquaculture industry (Edwards & Leung, 2009). In particular, few years ago the invasion by C. intestinalis in Canada was considered to be at “crisis level”, and the species has been considered a major marine invasive issue for the Department of Fisheries and Oceans of Canada (Edwards & Leung, 2009). Under these circumstances, understanding the ecological niche of this particular species complex would provide valuable information about how they manage to survive and establish dense populations in such distinct areas as the Mediterranean Sea and the much colder North Atlantic Ocean.

Along the temperate South American ecoregion, the species has been recorded from around 30°S Lat. in the Chilean coast, in the regions of Coquimbo and Antofagasta (Castilla et al., 2005), where it has been reported as being responsible for economic losses caused by damage to suspended cultures of Argopecten purpuratus (Uribe & Etchepare, 2002). According to Castilla & Neill (2009) the introduction and spread of the species into this region has been facilitated by the continuous transfer of propagules and materials between aquaculture centers. However, its wide physiological tolerance, reflected in its extensive world distribution (Madariaga et al., 2014), might facilitate the expansion of its range along the Chilean coast. Currently there are no official reports for the presence of C. intestinalis in southern regions (above 40°S), where most of the centers for the culture of molluscs are located (Norambuena & Gonzalez, 2005), and where small and medium size aquaculture centers play a key role in the economy and social interaction of local communities (Norambuena & Gonzalez, 2005).

A practical way to understand and ultimately predict range expansions of invaders is by characterizing the environmental conditions that are currently suitable for the persistence of a given species (Pearson, 2007), and then identifying those areas where such conditions are distributed in the geographic space (Colwell & Rangel, 2009; Franklin, 2010). A group of quantitative modeling approaches, known collectively as Ecological Niche Modeling (ENM) have been widely used for this purpose in recent years (Soberón & Peterson, 2005; Peterson, 2006; Soberón & Nakamura, 2009; Elith & Leathwick, 2009; Zimmermann et al., 2010; Peterson & Soberón, 2012). The central assumption of ENM is that the response functions estimated in these models provide an effective representation of the spatial response of the species to different environments (Cassini, 2011). In this sense, ENMs provide a quantitative and formal procedure to establish better plans of management and prevention through the assessment of risk or likelihood for potential or ongoing invasions (Locke & Hanson, 2009).

Despite their merits, the use of these methods in the management of invasive species requires two important difficulties to be overcome. First, results from ENMs rely heavily on the assumption that species are in equilibrium with the environment (Pearman et al., 2008; Colwell & Rangel, 2009; Peterson, 2011). To fulfill this main assumption it is important to take into account habitat similarity between ranges, aiming to ensure that the ENM analysis remains restricted to those areas that present similar environmental conditions (Randin et al., 2006). In this vein, analog environments in an invaded range represent those habitats inside the range of values considered to quantify the native range and so, correspond to environmental conditions that have been experienced by the species before the invasion; otherwise, habitat are considered non-analog, and conclusions about these environments must be taken with caution (Owens et al., 2013; Fitzpatrick & Hargrove, 2009). Second, results of ecological niche models (ENMs) are usually expressed as quantitative suitability indexes or as probability of presence, which are not necessarily linked to population abundances, a key parameter for pest managers or conservation biologists (VanDerWal et al., 2009). To overcome this second caveat, adequate modeling procedures and field validation of the fitted ENMs are necessary.

In this study we combine ENM with extensive field sampling to provide valuable information that might help the development of plans of control before these species become a significant problem. In particular we try to answer the following questions: Will the complex C. intestinalis continue to spread in the Chilean coast, or does it already occupy most of its potential range? If it continues to spread, will the spread extend to regions containing high concentration of aquaculture centers? Finally, are niche models indexes reflecting population abundances at a confidence level useful for pest managers and conservation biologists? The answers to these questions will provide key information for an adequate planning of prevention and control task in aquaculture centers, especially in southern Chile, where these centers represent a major economic activity for local communities.

Methods

Species occurrence data

Confirmed records on the occurrence of C. intestinalis were obtained from the Global Biodiversity Information Facility (GBIF— www.gbif.org). After removing duplicate records and a few records that presented obvious errors of georeference, the final dataset consisted of 776 from Northern Europe (considered here as the native range) as well as 107 presences registered in Canada, 98 records from the Pacific coast of the United States, 212 from Southern Europe, and 24 records from Japan. Due to their morphological similarity, and the consequent difficulty involved in the identification of each species in the field (but see Sato, Satoh & Bishop, 2012), we cannot unambiguously attribute records from the public databases to the different species within the genetc complex. Therefore, we decided to follow Zhan, Macisaac & Cristescu (2010) and allocate the set of occurrences to the species that dominates a specific area.Therefore, data from Canada and Northern Europe were considered as the current distribution of sp B. The other areas represent the distribution of sp A. We used occurrences from each area to calibrate single-models (hereafter called “Canada model” or “Southern Europe model”, etc.) and regional models (considering occurrences from more than one region where each species dominates). These single and regional models were then used to predict the potential distribution of C. intestinalis throughout the Chilean coast. Original distributions were defined using a 20 km buffer around the reported presence points.

Occurrences and density of C. intestinalis in the Chilean coast

To validate the results of our ENMs, we obtained confirmed presence of C. intestinalis by surveys in aquacultures centers throughout the Chilean coast. We visited the three main regions where aquacultures centers are located in northern, central and southern Chile according to the information provided by regional agencies of the Sernapesca (National Fishery Service). Centers producing oysters, mussels, abalones and scallops were visited. Fifteen localities were sampled from 27 to 43°S latitude along the Chilean coast (spanning approximately 1,700 Kms) during the summer seasons of the years 2013–2015. In each locality, all aquaculture centers and infrastructure (docks and pilings) were visually inspected for presence of C. intestinalis. Photographic records were taken and later were used to calculate the relative density of C. intestinalis in each site. Considering that some aquaculture centers are located closer than the resolution of oceanographic layers used in this study (see next section), we used average density by locality in our analysis to avoid pseudo-replication. Density was expressed as the number of individuals per 225 cm2 (15 × 15 cm grid).

Environmental variables

We chose oceanographic layers representing various quantitative environmental predictors with a recognized physiological and ecological relevance for C. intestinalis (Carver, Mallet & Vercaemer, 2006; Madariaga et al., 2014). These were Sea Surface Temperature—SST (minimum, mean, maximum and range), Photosynthetically Available Radiation—PAR (mean, maximum), Salinity (mean), pH (mean), Dissolved Oxygen—Dissox (mean), Chlorophyll A—Chlo (maximum, mean, minimum). All variables were obtained from BioOracle database (Tyberghein et al., 2012) with a spatial resolution of 5 arcmin (c. 9 km). Most of the grids contained monthly records for the period between 2002 to 2009, except PARmax and PARmean, which encompassed records from 1997 to 2009. The environmental layers were processed with Quantum GIS 2.6.0 to fit the extent of each zone.

Statistical methods

The dataset was separated into separate geographic areas to build single area models (East Canada, West USA, Japan, Southern Europe, Northern Europe), and regional models that grouped more than one area where each species dominates. For sp A we calibrated a model with records from Japan, United States and Southern Europe, and for sp B a model with occurrences from Northern Europe and Canada. We did not consider models for sp C and sp D, given their lower frequencies in empirical records and more restricted geographic distributions. We then used Niche Analyst (Qiao et al., 2014), to perform a Principal Component Analysis (PCA) on the environmental variables and visualize the environmental space into transformed principal component dimensions. The program uses a covariance-based approach to PCA calculation. We used minimum volume ellipsoids around the points of occurrence to delimit, in the environmental space, the conditions considered favorable for the persistence of the species. We later identified the geographic areas in the Chilean Coast where those environmental conditions can be found. Finally, we interpreted our results (suitability index) for those regions where the analog environments (similar conditions between the area where model was calibrated and the area of projection) were similar. This is relevant especially from a management perspective, because it makes easier to recognize areas with novel environments where niche model algorithms tend to extrapolate predictions. We also identified those areas most suitable for aquaculture in the Chilean coast (courtesy of Subsecretaria de Pesca–Subpesca). This allowed us to visualize the areas under higher risk of invasion and damage, and hence with more potential for economic losses.

Ecological niche models were fitted using Maximum Entropy Species Distribution Modelling software v.2.3 (Maxent). This is a useful method for making predictions especially when incomplete information about species distribution is available. By evaluating the climate data at each location where the species of interest is present, Maxent calculates a probability function that describes the chances of observing a presence giving the observed distribution of the species and the environmental conditions across the study area (Phillips, Dudík & Schapire, 2004; Phillips, Anderson & Schapire, 2006). The output of Maxent is a continuous variable which indicates environmental suitability. For each individual model, we used a 20-fold cross-validation scheme, except for the “Japan model”, where we used a 17-fold cross-validation scheme. The area under the curve (AUC) statistic for the Receiver Operating Characteristic (ROC) was used to measure how well each model discriminates presences more accurately than a random prediction (Phillips, Anderson & Schapire, 2006). Fitted models were later projected over the Chilean coast, using the same environmental variables, to identify where suitable environments for C. intestinalis are likely to occur. The importance of each environmental variable for every model was assessed by a jackknife procedure, fitting a model using each variable separately and ranking them according to the test gain.

Finally, we evaluate the relationship between suitability values with the observed density. To do so, we follow VanDerWal et al. (2009) and use linear regression and quantile regression (90% percentile) to determine if suitability indexes successfully predict the upper limit of local abundances. Goodness of fit was assessed using simple R2 values for linear regression, and pseudo R2 values (Koenker & Machado, 1999) for quantile regression.

Results

Principal component analysis of the pooled climatic data revealed three significant axes of climatic variation. The first three principal components accounted for 82.02% of the total variation in the data. The first principal component (PC1) was mainly thermal (related to SSTmean and SSTmin), whereas PC2 was related to SSTrange and the three Chlorophyll measures. Finally, PC3 was mainly related to salinity and pH. Interestingly, the PCA split the populations into two groups (Fig. 1). The first cluster includes mainly populations from Japan, Southern Europe and West USA. The other group embraces both, Northern Europe and populations from Canada. The climatic separation into both groups reflects the distribution of the two most common species that compose the genetic complex of C. intestinalis. Once the environmental space was defined using the first three principal components, we estimated the environmental niche by fitting a minimum volume ellipsoid around the presence points of each species. All points inside the ellipsoid represent environments with favorable conditions for the species in the original region. We projected the minimum volume ellipsoid in Chilean territory to identify analog environments (habitats inside the range of values found in the original region). Analog environments for sp A are almost four times more common in Chile than those for sp B according to the projections from the minimum volume ellipsoid (Fig. 2).

Figure 1 Principal component analysis for the environmental variables at the presence points.

PC1 and PC2 scores are shown. Colors represent different populations of C. intestinalis. Scores at PC1 markedly separate distributions of sp A (Japan, Southern Europe and USA) and sp B (Northern Europe and Canada).

Figure 2 Sampling points and projections of the potential distribution of C. intestinalis sp A and sp B on Chilean coast.

In the first map, red dots show the localities sampled into the three major aquacultural regions of Chile. Non analog environments are projections on Chilean environments that may be or may not be represented at locations used to fit the model. Analog environments are projections only on those Chilean environments that are represented at locations used to fit the model.

Models for original distributions were significantly better than random and performed well according to AUC (Table 1). The lowest AUC was obtained for the model of Northern Europe (AUC = 0.82), while the highest was obtained for the model of West USA (AUC = 0.94). For the regional models, the observed AUC values were 0.87 and 0.80 for sp A and sp B, respectively. Only regional models are shown. All the remaining models, obtained with single modelling areas as well as the projections with non-analog environments may be found in Figs. S1 and S2.

Table 1 Values of the average test AUC for the replicate runs for each single and regional model.

Environmental variables with the highest gain when used in isolation are shown.

Model	AUC	Variable with the highest gain	
Canada	0.834	Chlorophyll A (mean)	
Japan	0.889	Chlorophyll A (mean; maximum)	
West USA	0.939	Photosynthetically Available Radiation (PAR) (maximum)	
Southern Europe	0.892	Salinity; Ph	
Northern Europe	0.817	Sea Surface Temperature (minimum); PAR (mean)	
sp A	0.873	Salinity; Chlorophyll A (mean; minimum)	
sp B	0.804	Sea Surface Temperature (minimum); PAR (mean)	

The two regional models (sp A and sp B) showed strong disagreement for portions of Chile that are considered suitable for the establishment of C. intestinalis. Projections of the model of sp A showed that central and southern regions (30°−40°S, Fig. 2) seem more suitable for establishment than the peripheral regions of Chile (below 25° and above 43°S). Only a few patches of suitable habitats are found in the most extreme southern region of the continent, around the 53°S. The model for sp B revealed that the most suitable habitats are located around the 45°S (Fig. 2). Additional areas near 53°S were also classified with high suitability index, the same as observed with the model of sp A.

Based on Jacknife analyses for regional models the distribution of sp A was most influenced by mean Salinity and Chlorophyll A (both mean and minimum). For sp B, Sea Surface Temperature (minimum) and Photosynthetically Available Radiation (PAR) (mean) were important contributors (Table 1). The most important variables also varied in the single models, as in Canada where mean values of Chlorophyll A correspond to the variable that contributed most to the model, while in Southern Europe, Salinity and pH were the most important variables (Table 1).

Our field sampling detected C. intestinalis in most aquaculture centers examined in 15 localities between 27°and 43°S latitude (Fig. 2). To the best of our knowledge, this is the first report of C. intestinalis infestations south of 30°S latitude on the temperate South American pacific coast, specifically in mussel, oyster and abalone farms. Density of C.intestinalis showed a clear north-south pattern, with higher densities in northern Chile and low densities in the South.

The relative density of the species in aquaculture centers revealed contrasting results. For sp A. the plot showed a positive, linear relationship between observed density and the suitability index (R2 = 0.26, Fig. 3), and a strong relationship at the 90% percentile (pseudo R2 = 0.55, Fig. 3). For sp B, the relationship was completely absent, and for the upper limit was negative, which makes no sense in this context (R2 = 0.0, pseudo R2 = 0.08, Fig. 3).

Figure 3 Regressions between suitability indices from ENMs and observed densities in the field for sp A and sp B.

Dark lines represent linear regressions, gray lines represent 90% quantile regressions.

Discussion

Our results showed that, when considering the distribution of each species of the genetic complex of C. intestinalis (sp A and B), the species seem to be not only genetically distinct, but also appear to be ecologically distinct. Our analyses separated the species into two main groups based on their environmental preferences. Interestingly, the PCA analysis of environmental conditions for the presence points was coherent with the genetic separation among the species within the complex. In consequence, not considering these differences in environmental models might lead to imprecise conclusions about the potential distribution of the species outside their original range.

The first principal component is mostly influenced by mean and minimum values of sea surface temperature. These variables grouped occurrences from Northern Europe and Canada where the minimum sea temperature reach values close to 5 °C and separated them from observations obtained in areas like Japan, east USA and the Mediterranean Sea, where minimum values of temperature are much higher (11.7 °C, based on layers provided by BioOracle). Although the taxon is recognized by its wide tolerance to temperature variation (Carver, Mallet & Vercaemer, 2006), our results suggest a distinct range of preferences at least for the two most common species. By isolating the two groups, we could improve the power of our predictions by restricting the projections of the models to the geographic areas where the environmental conditions are analog to those where the models were calibrated. We took this precaution because some model algorithms tend to extrapolate projections beyond the range of environmental values used to calibrate the models and end up identifying high values of suitability even in conditions where most of the species are unlikely to survive (Owens et al., 2013).

The two regional models (for sp A and for sp B) showed disagreement for portions of Chile that are considered suitable for the maintenance of the species. Projections of the model of sp A showed that climate in Chile is moderately to highly suitable, especially around the 30° and 40°S, while for sp B, the areas around the 45°S are the most appropriate. Also, the model for sp B predicted several suitable areas beyond the extent of the current invaded range of C. intestinalis in the Chilean coast.

We interpreted projections in the Chilean coast only to those analog environments to those where the models were calibrated. For the model of sp A, it resulted in a void in the projection layer between the 34°–37°S and between 46°–51°S. These areas are known as strong upwelling centers, and they also receive important influxes of freshwater (Atkinson et al., 2002; Dávila, Figueroa & Müller, 2002). Both factors may generate particular conditions that are not shared by the other areas where the species has been found. In any case, our survey confirmed the presence of C. intestinalis in some aquaculture centers around the 36°S Lat. Initial colonization in this zone started, most likely, with specimens that benefited from the exchange of equipment between aquaculture centers from other parts of the country where the species has already established dense populations (IV Región, around 30°S). For sp B, analog environments, identified by the minimum volume ellipsoid, are just found southern 40°S, which means that only above this latitude coastal environmental conditions are similar to those the species found in Canada and Northern Europe.

The model for sp A seems to capture quite well the current distribution of C. intestinalis in Chile, especially in the area of Coquimbo (29°S 71°W) where the biofouling effect of the species has been a major problem on the suspended cultures of scallops (Uribe & Etchepare, 2002). Indeed, it is assumed the species first arrived in the area brought by Japanese boats which transported the personal and equipment used in the implementation of the first centers for the culture of scallop (Madariaga et al., 2014). The exchange of boats and equipment facilitated the spread of the species northward, where it can be found in dense populations also associated with cultures of scallops, especially during the summer. Later, the species has the potential to spread to the area close to Puerto Montt (around 41°S), although in a much lower density. Although the origin of the specimens found in the Chilean coast have been tentatively attributed to Japan, which could correspond to sp A, the single model adjusted with occurrences from Japan identified no analog environments in the Chilean coast (Figs. S1 and S2). This model is probably biased by the few points of occurrence that we obtained from the public registry for the area of Japan, and also to the proximity of the points. It means, our single model for Japan is probability characterizing only a narrow portion of the possible niche for the species. The model for sp B considered suitable areas situated outside the present distribution in Chile, and where most of the aquaculture centers are located. In some areas, the species is already established but still at low densities. Indeed, most of the mariculturalists that we could contact in southern regions (above 40°S) did not recognize C. intestinalis as a real threat to their cultures, which contrasted with the response from mariculturalists from the northern-central Chile, who could readily identify C. intestinalis and view it as a real threat. Anecdotally, farm workers only recognize C. intestinalis correctly in the northern region of Chile. In the farms located at southern Chile, workers misidentify C. intestinalis with early stages of Pyura chilensis. Hence, southern regions must be considered a priority in future plans of management and control, which should include programs to provide adequate training to local mariculturalists to improve the changes of early detection of new invasions and to prevent contamination due sharing equipment between farms.

Madariaga et al. (2014) used information from unifactorial experiments to assess the tolerance to light, salinity and temperature of individuals collected in the Region of Coquimbo (30°S). They also compiled data from literature to associate performance (mixing several metrics as mortality, filtration rate and particle retention efficiency) with salinity and temperature. These authors suggest that the species is physiologically capable of tolerating the wide range of physical conditions found in Chile, suggesting the whole Chilean coastline may be considered at high risk. However, data compiled in this study was not separated into the different species within the complex (i.e., sp A, B, C or D); therefore, the observed performance cannot be assigned to any species in particular or be considered representative of the species already present in Chile. To the best of our knowledge, there are no available studies that compare the physiological tolerance of each of these four species independently. Nevertheless, our results suggest the two dominant species of the genetic complex are probably using different portions of the environmental space.

The Jacknife analyses that included each variable alone, reinforced the separation of the two species in the environmental space. For sp A, Salinity and Chlorophyll A (mean; minimum) are the most important variables. For sp B, Sea Surface Temperature (minimum) and Photosynthetically Available Radiation (PAR) were important contributors. Previous studies have reported that temperature is an important cue for sexual maturation, spawning and recruitment in C. intestinalis (Dybern, 1967; Marin et al., 1987; Carver, Chisholm & Mallet, 2003; Howes et al., 2007). For instance, in Scandinavian and subarctic populations, where temperature rarely exceeds 8 °C, the generational time is 2–3 years and individuals are reproductively mature at the first year (Dybern, 1965; Dybern, 1967). For Japan and the warmer Mediterranean where temperatures are always above 10 °C, the generational times vary between 3 and 6 months and sexual maturity is reached after 1–2 months, depending on the season (Yamaguchi, 1975). The optimal salinity for Mediterranean populations (35‰), is much higher than would normally experienced by northern Atlantic coastal populations (Marin et al., 1987). Lambert & Lambert (1998) reported that C. intestinalis populations on floating docks in southern California harbors were vulnerable to pulses of low salinity. On the other hand, Dybern (1967) found that the lower salinity limit for adults and developmental stages in Scandinavian populations was 11‰. Such differences might emphasize the capability of the species to survive under a broad range of conditions, but can also reinforce the implications from our results, that each species of the genetic complex might be using different portions of the niche. This information is crucial, especially when using ENMs to study how species colonize new environments (Sax et al., 2007) and whether they retain their climatic niche in a new range (Pearman et al., 2008).

Here, we observed a linear relationship between relative density and suitability index for sp A, especially in relation to the upper limit of local abundances (Fig. 3, 90% quantile regressions). VanDerWal et al. (2009) pointed out that suitability indexes reflect potential abundance, but other factors may prevent the species attain this potential. In our case, the index explains 56% of the variation on the upper limit of the population density among aquacultures centers, which means that those centers located in areas classified as possessing a high suitability index could sustain more abundant populations. For sp B, the same relationship was null. In our case, we can speculate the reason why some localities do not show the expected higher abundance could be associated to environmental variables outside the model or acting at a small scale such as turbidity or management activities in aquaculture centers; however, finding the exact reason will require new studies.

According to the National Fishery Service, the areas around the 42°S encompass most of the centers for the culture of molluscs in Chile. Such areas must be of high priority for control plans. Some areas above the 50°S may also serve as potential habitat for C. intestinalis from a strictly climatic perspective. However, they are not considered suitable for aquaculture, so it is not clear whether they could support wild population of the species. In fact, Dumont et al. (2009) suggest that despite the well-established populations on artificial structures, the species appears unable to colonize natural communities due to predation pressure from native benthic species, especially the rock shrimp Rhyncocinetes typus. On the other hand, competition with the native ascidian Pyura chilensis seems unimportant in natural benthic communities exposed to natural predation (Dumont et al., 2009).

In the case of the few places where marine invaders have been successfully controlled, first actions typically occurred in the early stages of invasion, right after establishment and initial spread (Edwards & Leung, 2009). The higher a population size, the longer the species will persist, and eradication will be no longer a management option (Lockwood, Hoopes & Marchetti, 2013). In this context, our results might help the Chilean regulatory agencies to identify which areas must be prioritized in eventual control plans. Considering the potential risk to southern Chile, management of C. intestinalis invasions should concentrate on the reduction of the probability of introduction due to contaminated individual vessels (Drake & Lodge, 2004), controlling the number of potential invaders on transport or recreational boats. Controlling the exchange of contaminated equipment between aquaculture centers might also reduce the fortuitous spread of the species.

Risk maps are in worldwide demand for management purposes, however they are clearly dependent on the type of occurrence data used (Therriault & Herborg, 2008). If it is possible to link suitability and abundance, ENMs may turn into a very powerful tool in the management of invasive species. Even if management measures have not been able to eliminate biological invasions, slowing the rate of invasion or spread of an established species has considerable value (Ruiz & Carlton, 2003). In our case, ENMs have allowed us to understand the relation of the distinct species that conform the genetic complex known as C. intestinalis with their surrounding environment providing valuable insights about probable routes of dispersion in Chile, especially into those areas considered adequate for aquaculture activities and where the species has not been recorded.

Supplemental Information

Figure S1 Non-analog maps

Projections of the potential distribution of C. intestinalis on Chilean coast using non analog environments for each of the five locations used in the analysis.

Click here for additional data file.

Figure S2 Analog maps

Projections of the potential distribution of C. intestinalis on Chilean coast using just analog environments for each of the five locations used in the analysis.

Click here for additional data file.

We thank Daniela Lopez, Melissa Pavez, Kennia Morales, Roger Sepúlveda for their valuable help during field work, and to all anonymous mariculturalists that kindly received us and shared their experience with the field team. This paper was much improved by the comments of Dr. AT Peterson.

Additional Information and Declarations

Competing Interests

Author Contributions

Data Availability

The authors declare there are no competing interests.

Stella M. Januario conceived and designed the experiments, performed the experiments, analyzed the data, contributed reagents/materials/analysis tools, wrote the paper, prepared figures and/or tables.

Sergio A. Estay conceived and designed the experiments, analyzed the data, contributed reagents/materials/analysis tools, wrote the paper, prepared figures and/or tables, reviewed drafts of the paper.

Fabio A. Labra wrote the paper, reviewed drafts of the paper.

Mauricio Lima reviewed drafts of the paper.

The following information was supplied regarding data availability:

Global Biodiversit Information Facility (GBIF): http://www.gbif.org/occurrence/search?taxon_key=2329622&HAS_COORDINATE=true&HAS_GEOSPATIAL_ISSUE=false.

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
