# Peer review of "Combining environmental suitability and population abundances to evaluate the invasive potential of the tunicate Ciona intestinalis along the temperate South American coast"

_PeerJ, doi:10.7717/peerj.1357_

## Round 0.1 · original submission · Minor Revisions

· Academic Editor

Minor Revisions

Dear authors

Thank you for submitting your manuscript to our journal. I am sorry about the delay in sending you the decision, but it was difficult to secure two reviewers. As you see both reviewers liked the science and the manuscript and suggest minor revisions of your ms. Please pay attention as reviewers have provided feedback as text as well as via annotations on the manuscript. In addition, feedback has been left on your preprint at https://peerj.com/preprints/1281v1/#feedback

Do address all comments individually. If you are willing to do so, we would be happy to reconsider your revised manuscript.

·

Basic reporting

No comments

Experimental design

No comments

Validity of the findings

No comments

Comments for the author

This is an interesting piece of work which goes beyond the standard reporting of invasive species distributions to attempt to predict likely future distributions along a long latitudinal gradient (the Chilean coast). I have made some comments on an annotated pdf but they are all fairly minor. It would be worth you seeking to shorten both the abstract and discussions. I also wonder if you should make reference to the potential of this approach for this species in predicting likely impacts of climate change on its future distribution?

·

Basic reporting

The ms provided abides with the PeerJ policies and conforms to the rules about the structure of the article. The article is concisely written, with one or two minor points of clarification necessary. It is very clear how the article informs and adds to the field, and sufficient background material is provided in order to show broader relevance.

The figures are relevant (but another might be helpful - see additional comments below), and provide valuable information for the article.

I have highlighted some references that do not appear to be cited in the text and the reference list needs to be checked to ensure they are all in the same format.

Experimental design

Whilst I am not very familiar with the use of ENM's (nor the computation involved in their application), the paper appears to report a well-designed test of and application of such techniques to modelling the abundance of an invasive marine species. Public sourced data is used to generate a model of possible distributions of the taxa and sampling conducted in the field to determine if the taxa were found in the locations highlighted as possibly suitable for colonisation. One or two minor points of clarification would aid the non-specialist in the understanding of the findings, but overall the research questions are clearly defined, tested and discussed.

Validity of the findings

Again, I am not that familiar with the use of these models and their application. The methods applied appear to be robust in concert with the literature cited, and the conclusions are appropriate.

Comments for the author

This is a very interesting paper, illustrating the use of ENM's to model the possible distribution of a marine species and coupled with empirical evidence to test the outcomes of the model. It should certainly be published in PeerJ (as it is an interesting subject and certainly adds to the body of knowledge in the area), and forms an elegant piece of work. I have highlighted on the ms some minor alterations to the text and highlighted a couple of areas where I personally needed a little more clarification as a non-subject specialist in this modelling technique.

---

## Round 0.2 · accepted · Accept

· Academic Editor

Accept

Thank you for your revision of the text. I am now satisfied, as are the referees, that you have addressed the issues raised, and am happy to recommend your manuscript to move into publication.

·

Basic reporting

Fine

Experimental design

As previously

Validity of the findings

As previously

Comments for the author

The corrections and amendments made are sufficient for this paper to be published.

·

Basic reporting

No further comments

Experimental design

No further comments

Validity of the findings

No further comments

Comments for the author

The paper has been revised by the authors and they have responded to all the comments made by the reviewers. I am pleased to recommend acceptance of the revised ms.